# Possibilities of Using the Duplex System Plasma Nitriding + CrN Coating for Special Components

David Dobrocky [1], Zdenek Pokorny [1,*], Roman Vitek [2], Jiri Prochazka [1], Zbynek Studeny [1], Zdenek Joska [1], Josef Sedlak [3], Martin Slany [3] and Stepan Kolomy [3]

1  Department of Mechanical Engineering, Faculty of Military Technology, University of Defence, 662 10 Brno, Czech Republic
2  Department of Weapons and Ammunition, Faculty of Military Technology, University of Defence, 662 10 Brno, Czech Republic
3  Department of Machining Technology, Faculty of Mechanical Engineering, Brno University of Technology, 616 69 Brno, Czech Republic
*  Correspondence: zdenek.pokorny@unob.cz; Tel.: +420-973-442-839

**Abstract:** The article deals with the replacement of hard chrome plating by applying the duplex system plasma nitriding + CrN coating (hereinafter referred to as PN + CrN). The goal of the research was to find a suitable alternative for steel surface treatment that would replace hard chrome plating and ensure similar mechanical and tribological properties. An exposed part of a small-bore weapon was selected for evaluation, namely the gas piston of the 42CrMo4 steel breech mechanism drive. The PN + CrN duplex system was compared with a hard chrome coating as well as a self-deposited CrN coating. The mentioned surface treatments were evaluated in terms of metallography, mechanical and tribological properties and surface texture. From the mechanical properties, the hardness of the surface was analyzed, an indentation test was performed (Mercedes test) and adhesive-cohesive behavior was evaluated (Scratch test). Furthermore, an instrumented penetration test was performed (an evaluation of plastic and elastic deformation work and indentation hardness). As part of the assessment of tribological properties, the Ball-on-Flat test, the measurement of the coefficient of friction and the measurement of traces of wear were performed. The surface texture was evaluated in terms of morphology and surface roughness measurement by selected 2D and 3D parameters. The PN + CrN duplex system showed higher hardness than hard chrome, better tribological properties (lower friction coefficient), but worse surface texture. The PN + CrN duplex system has proven to be a suitable alternative to the hard chrome coating for exposed parts of small-caliber weapons, which can be applied in arms production.

**Keywords:** plasma nitriding; hard chrome; CrN; duplex system; weapon





## 1. Introduction

A shot from a weapon is a method of accelerating a projectile in the barrel to the appropriate initial velocity by employing the pressure of gases produced by the combustion of the propellant charge [1]. The operating cycle of a weapon refers to the series of activities done between two successive rounds.

A drive by absorbing propellant gas is a fairly common propulsion, especially for higher-powered small arms weapons (submachine guns, assault rifles, and machine guns). The main functional part of the weapon systems is set in motion utilizing a pulse of pressure of the propellant gases taken during the shot from the bore of the barrel in this method of propulsion.

Only the first of them, piston-driven propulsion, is currently employed exclusively for automatic guns. A piston mechanism of the Czech automatic rifle 7.62 mm Sa model 58, which uses a 7.62 × 39 mm cartridge, is an example of this method of propulsion (7.62 mm model 43). The main parts of this device are (see Figure 1):

- gas port;

- gas cylinder;
- gas piston with piston rod;
- gas channel, which connects the bore of barrel with the gas cylinder.

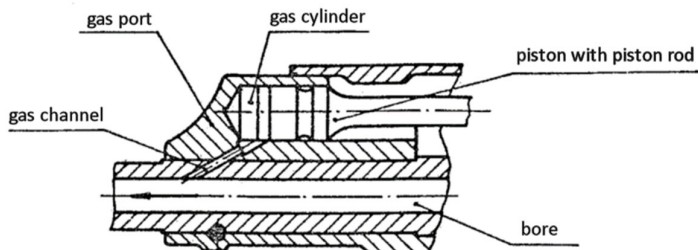

**Figure 1.** Piston system at automatic rifle Sa model 58. [1].

During the shot, when the projectile's bottom passes through the gas channel, a portion of the propellant gases penetrates the gas cylinder, causing a rapid increase in pressure $p$ and temperature $T$ in the gas cylinder's volume $V$. The force $F$ created by the pressure of the propellant gases in the gas cylinder acting on the front surface of the gas piston $S$ accelerates the piston and piston rod to the maximum speed $v_{max}$. The force $F$ is conveyed to the main functional part of the weapon mechanisms, which is the breech block carrier in the case of the automatic rifle Sa model 58. The kinetic energy of the breech block carrier is then used to drive the other mechanisms of this weapon.

During the shot, there is an intense strain on individual parts of the automatic firearm, including parts of the piston system. These are mainly the following types of stress:

- pressure action of propellant gases;
- temperature action of propellant gases;
- mechanical interaction between moving parts of the weapon (especially dynamic shocks).

One of the ways to increase the service life of exposed weapon components, such as gas pistons, barrel bores, parts of the breech mechanism (locking piece, breech block, breech block carrier, etc.), is the application of suitable coatings. Due to their suitable chemical and physical properties, chromium coatings are mainly used, created by the process of hard chromium plating (sometimes referred to as functional chromium plating) [2–4]. Hard chromium plating is an electrochemical process used to deposit a layer of chromium on a substrate. It is used in applications where high hardness and abrasion resistance or prolonging the service life of the functional surfaces of components is required. These coatings are formed in thicknesses of 10 µm to 100 µm. The hard chrome plating process itself has both advantages and disadvantages. The advantages, such as relatively low acquisition costs, have recently been overshadowed by a number of disadvantages, including low current efficiency, low resistance to chlorides, sulphuric acids and tensile residual stresses, causing lower corrosion resistance and reduced fatigue strength [5,6]. However, one of the biggest disadvantages is certainly the unecological nature of the whole process. During the coating process, compounds containing hexavalent chromium are present, which is very dangerous and one of the substances with the highest potential to cause cancer [7–9]. Furthermore, it is necessary to prepare acid baths, which again represents a significant environmental burden in the entire process.

Currently, the trend is to find suitable replacements for hard chrome plating technology, which is still used for a wide range of applications. These substitutes include, for example, the deposition of hard, abrasion-resistant coatings using PVD (Physical Vapor Deposition) and CVD (Chemical Vapor Deposition) technologies. For example, the application of PVD coatings based on FeinAl (designation of the AlCrN coating) showed that the application of this coating to the gas piston did not lead to its visible wear even after 3000 shots [10]. However, it should be noted that the FeinAl coating was applied to the original hard chrome coating. Another possible substitute for hard chrome plating is the use of thermal spraying, e.g., HVOF (High Velocity Oxygen Fuel) technology [11–14]. However, HVOF technology

is not suitable for all applications, as the coatings created by this technology must in many cases be further processed, e.g., by grinding or polishing [15,16]. Another disadvantage of this method is the porosity of the formed coating, although it is significantly lower compared to other thermal spraying methods (e.g., TWAS—Twin Wire Arc Spraying, APS—Atmospheric Plasma Spraying) [17–19]. Good properties of chrome coatings have been achieved using the HVAF (High Velocity Air-Fuel) method [20–24], which can also be used as an alternative to hard chrome plating. Currently, CS (Cold Spray) technology is coming to the fore, which can be used to replace hard chrome with alternative surface processes or to produce a hard chrome layer without the use of Cr6 [25–27]. Currently, the possibility of replacing exposed steel components with composite materials is increasing [28–30].

Chromium nitride (CrN) coated by Physical Vapor Deposition (PVD) has proven to be a promising replacement for hard chromium plating due to its high hardness, low internal stress, toughness and ability to improve corrosion, oxidation and wear resistance [31,32]. The useful properties of this coating can be increased by chemical-heat treatment of the base material (substrate), i.e., by creating a hard layer-coating duplex system [33–35]. The diffusion layer formed, for example, by the nitriding process, increases the hardness of the substrate surface, which improves the adhesion of hard coatings on soft substrates and further increases the load-bearing capacity and fatigue strength of the substrate [36]. In general, with this technology, great emphasis is placed on the preparation of the functional surfaces of the components before coating. The surface texture achieved by the production technology is largely copied to the texture of the coating (so-called technological inheritance), so it is appropriate to use a suitable finishing method (e.g., polishing, sandblasting, etc.) [37], before applying the coating (so-called surface pretreatment) or suitably finishing the surface of the deposited coating (e.g., wet sandblasting, dry sandblasting, etc.).

In this work, selected mechanical and tribological properties of CrN coating deposited by PVD technology on a substrate in the form of 42CrMo4 steel, which is widely used for the production of exposed weapon components, including gas pistons, were compared. The properties of both the CrN coating itself and the duplex system plasma nitriding + deposition of the CrN coating (PN + CrN) were investigated. The results of the investigation of the properties of the CrN coating and the PN + CrN duplex system were compared with the results of the hard chromium coating formed by a standard electrochemical process. Hard chromium was deposited on 42CrMo4 steel in the tempered state. The aim of the study was to investigate the possibility of replacing hard chromium with a CrN coating or a PN + CrN duplex system.

## 2. Materials and Methods

### 2.1. Determination of Stress Magnitude

In terms of absolute values of individual types of stress, the pressure and temperature action of propellant gases is dominant. Depending on the type, caliber and ballistic power of the weapon and the type of gunpowder used, the maximum values of propellant gas pressure in the bore are mainly in the range of 100–600 MPa. In extreme cases, propellant gas pressures can reach up to 1 GPa, while the maximum propellant gas temperature reaches values in the range 2200–3800 K during the shot [38]. The determination of propellant gas pressure values is possible numerically and experimentally.

Numerical determination of pressure values is the subject of the theory of interior ballistics of firearms [38]. The courses of interior ballistic characteristics (pressure and temperature of propellant gases, trajectory and velocity of the projectile as a function of time) are determined by solving a system of interior ballistics equations, derived on the basis of thermomechanical theory of ideal gas and Newton's laws of motion.

Experimental determination of propellant gas pressure values is possible using pressure gauges or piezoelectric pressure sensors, either by measuring directly in the initial combustion chamber using insertion pressure gauges (in the case of artillery weapons), or by measuring on special ballistic barrels using threaded pressure gauges [38]. Experimental determination of the propellant gas temperature during the shot is very problematic and

in fact practically impossible due to the small measuring ranges of available temperature sensors (up to 2000 K for thermocouples type B and D) and long sensor response times, which fluctuates for thermocouples and thermistors in the range of 0.1–1 s [39]. Thus, only the maximum, so-called explosion temperature $T_v$ of a given propellant charge is determined experimentally, indirectly by calculation from the experimentally determined explosion heat $Q_v$ by burning a certain amount of propellant charge in a calorimetric test pressure vessel immersed in a calorimeter. The course of the propellant gas temperature as a function of time is then determined numerically within the solution of the system of interior ballistics equations for the periods of combustion and expansion of propellant gases [38].

In current design practice, the determination of the pressure and temperature profiles of the propellant gases in the gas cylinder of a piston gas device is carried out only numerically. Experimental determination of propellant gas pressure is not possible due to the requirements for the installation of modern propellant gas pressure sensors, which in the case of piston gas devices cannot be met due to their relatively small dimensions. To determine the course of pressure and temperature in the gas cylinder, the calculation method of prof. Popelínský [40], based on the application of the 1st law of thermodynamics for an open thermodynamic system (energy conservation law) in the form [41] is employed, namely:

$$\frac{dQ}{dt} + \sum_j \frac{dH_j}{dt} = \frac{dU}{dt} + \frac{dA}{dt},\qquad(1)$$

where $\frac{dQ}{dt}$ is change of heat supplied, respectively discharged from the gas cylinder, $\sum_j \frac{dH_j}{dt}$ is the sum of changes in the enthalpies of the gases supplied, respectively discharged from the gas cylinder, $\frac{dU}{dt}$. is change of internal gas energy in a gas cylinder and $\frac{dA}{dt}$. is change in the volumetric work performed by the gas in the gas cylinder.

Furthermore, the equation of state in a gas cylinder applies in the form of:

$$p \times v = r \times T,\qquad(2)$$

where $p$ is gas pressure in the gas cylinder, $v$. is specific volume of gas in the gas cylinder, $r$ is specific gas constant of a gas in a gas cylinder and $T$ is gas temperature in the gas cylinder.

Furthermore, Newton's second law of motion is used, applied in the form of the equation of motion of the piston together with the piston rod and the main functional member (hereinafter MFM) in the form:

$$m \times \frac{d^2x}{dt^2} = p \times S,\qquad(3)$$

where $x$ is the path of the piston together with the piston rod and MFM, $m$ is the weight of the piston together with the piston rod and MFM, $p$ is gas pressure in the gas cylinder and $S$ is forehead surface of the piston.

Using the law of energy conservation, the equation of state and the equations of motion of the piston together with the piston rod and the MFM, it is possible to determine the time courses of the pressure $p$ and the temperature $T$ acting on the piston. The propellant gas pressure was calculated for a 7.62 × 39 mm cartridge (Figure 2a).

The results of the calculation were compared with the measurement (differences in the waveforms are caused by the fact that the result of the calculation is the so-called mean ballistic pressure, while the measurement found the course of pressure as a function of time for the cartridge mouth).

The calculated course of pressure and temperature of propellant gases was then used to determine the course of pressure and temperature of propellant gases in the gas cylinder, with the fact that in the energy conservation equation a simplistic assumption was made that there is no heat exchange with the environment, i.e., $\frac{dQ}{dt} = 0$. The calculation results

for real design parameters of automatic rifle Sa model 58 are shown in the following graphs (Figure 2b,c).

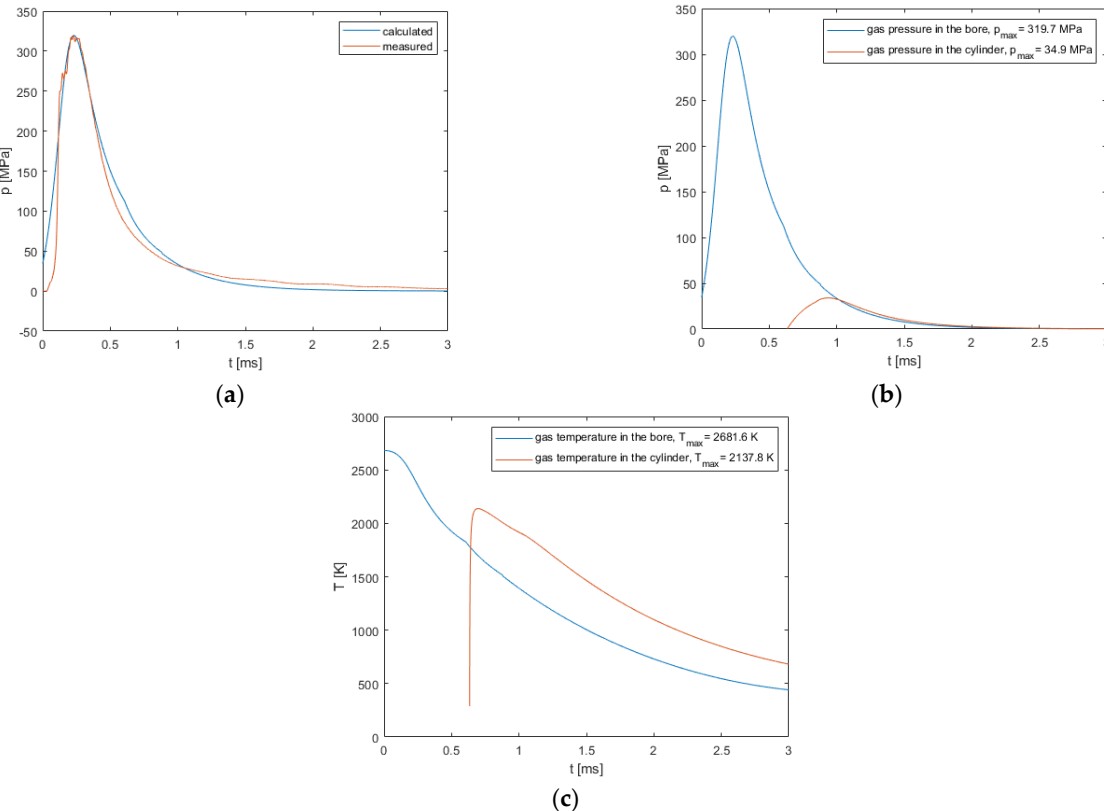

**Figure 2.** (**a**) The comparison of the calculated and measured course of propellant gas pressure as a function of time for cartridge of 7.62 × 39 mm; (**b**) the comparison of pressure; (**c**) the temperature in the bore of the barrel and in the gas cylinder.

Based on the above results, it is clear that the gas piston must withstand high stress during operation, the piston forehead is exposed to dynamic shocks, high temperatures and pressures. The piston surface must withstand friction and wear, corrosion and erosion, and oxidation [42–44]. Failure of the piston function leads to limited operation or malfunction of the weapon.

### 2.2. Materials

Disk-shaped specimens with a diameter of 70 mm and a thickness of 6.6 mm were produced from 42CrMo4 steel. Functional surfaces were made by circumferential surface grinding on a BPH300 (TOS, Hostivař, Czech Republic) surface grinder, 250 × 25 × 76 wheel, with a grit of 60 and a hardness of J. The roughness Ra after grinding was achieved in the range of 0.50 to 0.70 μm. The samples thus produced were refined to a yield strength value corresponding to a tempering temperature of 600 °C. Hardening was performed at 840 °C, with an austenitization temperature duration of 15 min, into water; tempering at a temperature of 600 °C, with a duration of 60 min, again into water. To suppress oxidation and decarburization of the sample surface, Kalsen 3 (AZ Prokal, Brno, Czech Republic) protective coating was used during heat treatment. The treated samples were re-ground with a 250 × 25 × 76 wheel, with a grain size of 80–180 and a hardness of K. The resulting roughness Ra was achieved in the range 0.30–0.35 μm. A total of nine samples were produced, three for each surface treatment application (hard chrome, CrN coating and PN + CrN duplex system).

### 2.3. Chemical-Heat Treatment and Deposition

Prior to the chemical-heat treatment process and the deposition of the coatings, the samples were cleaned by ultrasound in an ethanol bath for 15 min.

Plasma nitriding was performed on a Rübig PN 60/60 device (Rübig Group, Wels, Austria). Prior to nitriding, the surface of the samples was pre-cleaned in an atmosphere ($H_2 + N_2 + Ar$), at a pressure of 80 Pa, for 45 min, at a temperature of 530 °C and a voltage of 700 V. The plasma nitriding process itself took place at a temperature of 540 °C, for 6 h, in an atmosphere of 90 $L \cdot h^{-1}$ $N_2$, 30 $L \cdot h^{-1}$ $H_2$, 4.5 $L \cdot h^{-1}$ $CH_4$, pressure 280 Pa, voltage 520 V.

The chromium coating of selected samples was formed in a bath with a composition of 220 $g \cdot L^{-1}$ $CrO_3$ + 2.2 $g \cdot L^{-1}$ $H_2SO_4$, at a temperature of 60 °C and a process time of 35 min; the current density was set at 30 $A \cdot dm^2$.

CrN coating deposition was performed on a π1000 Platit (BCI Blösh Group, Grenchen, Switzerland) at 430 °C for 6 h, 100 V bias, 160 A arc and $N_2$ pressure 0.01 mbar, Cr target.

The parameters of the chemical-heat treatment process were determined on the basis of long-term research on diffusion technologies at our workplace. Such parameters were chosen that are used on highly-stressed components of special equipment (gear wheels, shafts, ball screws and weapon components). The chrome coating parameters were chosen based on the procedures established for chrome plating the gas pistons of the evaluated weapon during their manufacture. The CrN coating deposition parameters were chosen in cooperation with the coating equipment manufacturer.

The configuration of the individual surface treatments is shown in Figure 3. Table 1 documents the basic selected properties of the individual materials used in the proposed configurations [45,46].

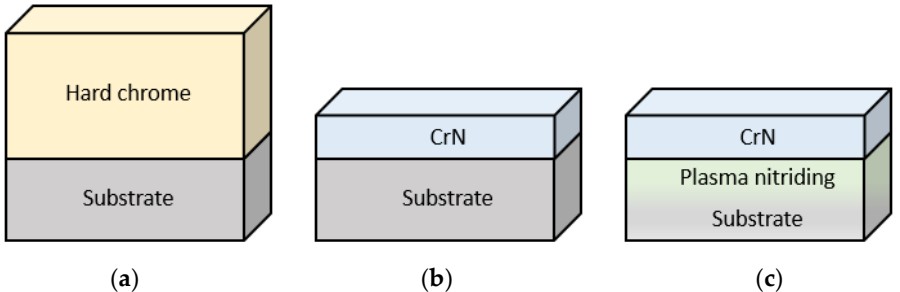

**Figure 3.** Configuration of evaluated surface treatments: (**a**) hard chrome; (**b**) CrN coating; (**c**) PN + CrN duplex system.

**Table 1.** Basic selected properties of used materials.

| Material | Hardness HV | Density ($g \cdot cm^3$) | El. Resistance ($\mu\Omega \cdot cm$) | Coeff. of Thermal Expansion (μm/m/K) | Tensile Strength ($N/mm^2$) |
|---|---|---|---|---|---|
| 42CrMo4 steel | 280–330 | 7.85 | 19 | 11.1 | 981–1177 |
| Hard Chrome | 575 | 8.10 | 55 | 13 | 850 |
| CrN | 2800 | 5.9 | 14,900 | 2.3 | 203 |
| Nitriding layer | 850–950 | - | - | - | - |

### 2.4. Characterization

The chemical composition of the 42CrMo4 steel was evaluated on a Q4 Tasman spark optical emission spectrometer (Bruker, Billerica, MA, USA). The surface hardness of ground samples, samples after plasma nitriding, samples with coating of hard chrome, CrN and duplex system PN + CrN, was measured on a Zwick ZHU 2.5 device (Zwick Roell Group, Ulm, Germany), load 1 kg, in accordance with ISO 14577-1:2015 [47]. The same equipment also performed an indentation test of adhesion, the so-called Mercedes test, which is defined by the VDI 3198 standard [48]. The scratch adhesion test, according to ASTM C1624-05 [49], was performed using a Bruker UMT-3 TriboLab tribometer (Bruker, Billerica, MA, USA),

under a load of 20 N–80 N. The Ball-on-Flat method, which is defined by the ASTM G133-95 standard, was used to analyze the tribological properties [50]. The load was chosen to be 10 N, the frequency 5 Hz and the test time 1000 s. The test was performed again on a Bruker UMT-3 TriboLab (Bruker, Billerica, MA, USA). The Olympus DSX500i opto-digital inverted microscope (Olympus, Shinjuku, Japan) at 2000× magnification and the Tescan MIRA 4 analytical scanning microscope (Tescan, Brno, Czech Republic), with SE detector and field of view (FoV) 20 μm, working distance (WD) 4 mm and accelerating voltage 15 keV, were used for image analysis of the surface, wear marks and metallographic structure. Surface roughness evaluation was performed on a Talysurf CCI Lite coherence correlation interferometer (Taylor Hobson, Leicester, UK), on an evaluated area of 0.8 mm × 0.8 mm; the evaluated length for measuring 2D parameters was 4 mm, cut-off 0.8 mm, a Gaussian filter was used to filter the data. For metallographic evaluation of the samples, transverse cuts were made, which were then pressed into the thermoplastic material. The samples were ground on a Leco PX 500 universal polisher (Leco, St. Joseph, MI, USA) using grinding wheels PT1, PT3 and PT4 (Leco, St. Joseph, MI, USA). The samples were polished on velvet with a 0.5 μm diamond-grained abrasive paste. Sample etching was performed with Nital 2% to 4%. The thicknesses of the formed layers were measured optically, the depths of the diffusion layers were measured using an automated microhardness tester AMH55 (Leco, St. Joseph, MI, USA) using three microhardness curves, on polished samples.

## 3. Results

### 3.1. Chemical Composition and Metallography

The chemical composition of the 42CrMo4 steel is documented in Table 2. The composition of the steel corresponds to the standard values given in EN ISO 683-2: 2018 [51].

**Table 2.** Chemical composition of 42CrMo4 (in wt. %).

| C | Mn | Si | Cr | Ni | Mo | P | S | Fe |
|---|---|---|---|---|---|---|---|---|
| | | | | Q4 Tasman | | | | |
| 0.42 | 0.66 | 0.29 | 1.09 | 0.12 | 0.16 | 0.002 | 0.003 | rest |
| | | | | Standard | | | | |
| 0.38–0.45 | 0.50–0.80 | 0.17–0.37 | 0.90–1.20 | max. 0.50 | 0.15–0.30 | max. 0.035 | max. 0.035 | rest |

The basic (tempered) state of 42CrMo4 steel consisted of the structure of tempered martensite and sorbite. The same structure is visible in the sample after the chromium plating process (Figure 4a). The continuous layer of chromium on the steel surface reached a thickness of 8.5–9.0 μm.

The structure of the steel after the PVD deposition process of the CrN coating is documented in Figure 4b. The structure shows tempered martensite with a higher proportion of sorbite, which corresponds to a "secondary" tempering during coating deposition. The thickness of the CrN coating is in the range 2–3 μm.

After the plasma nitriding process, a nitrided layer with a depth of $287 \pm 27$ μm was formed (Figure 4c).

On the steel surface, a very thin purely nitride region (compound layer) is visible, formed by nitrides, respectively, carbonitrides of type $\varepsilon$ ($Fe_{2-3}N$) and $\gamma'$ ($Fe_4N$) of iron and alloying elements. Its structure is influenced by the technology of the saturating process and the composition of the steel. Below the compound layer is a diffusion layer, which consists of ferrite (nitrogen ferrite) and nitrides (carbonitrides) of Fe and alloying elements. Nitride formation is essentially a precipitation process from nitrogen supersaturated ferrite. The highlighted boundaries of the original austenite grains are visible in the diffusion layer. This phenomenon can be attributed to the increased diffusivity of the elements at the grain boundaries and thus to their higher nitrogen saturation after plasma nitriding. The core structure consists of tempered martensite and sorbite.

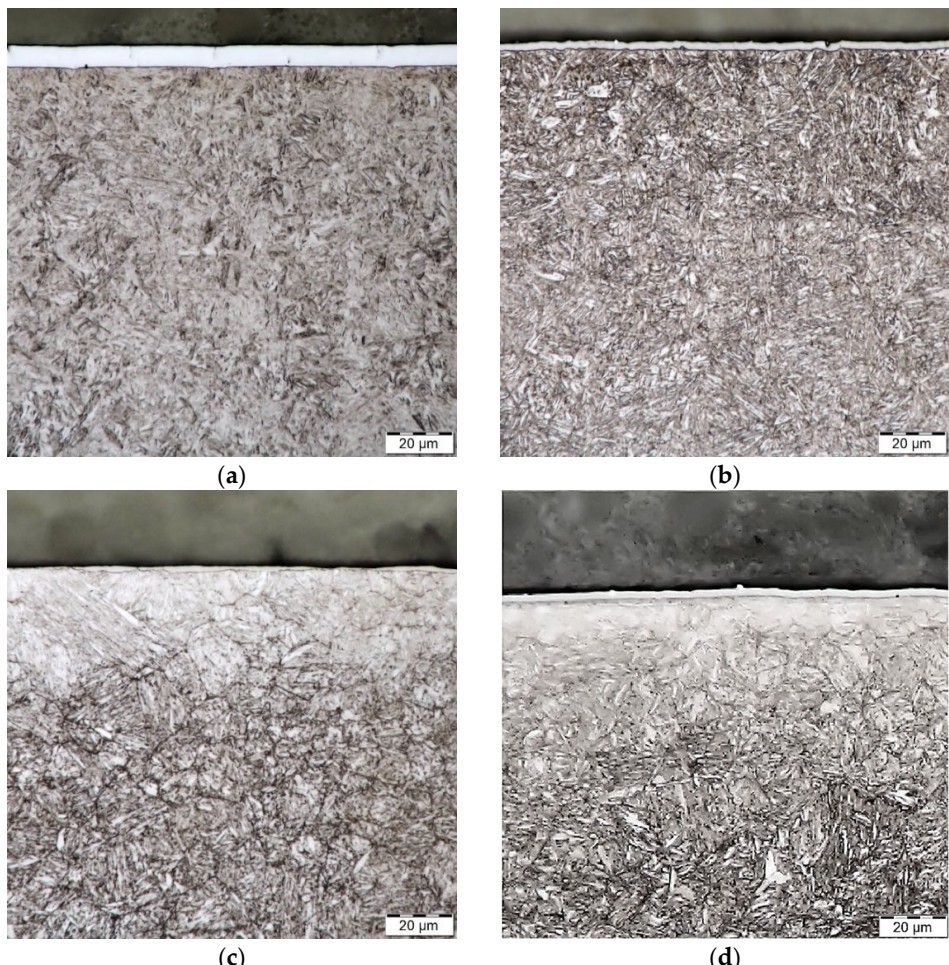

**Figure 4.** Metallography of 42CrMo4 steel: (**a**) hard chrome; (**b**) CrN; (**c**) plasma nitriding; (**d**) PN + CrN.

The metallographic structure of the PN + CrN duplex system is documented in Figure 4d. The structure of the base material (core) is similar to the structure of steel after plasma nitriding, i.e., it consists of tempered martensite and sorbite. The original boundaries of the austenitic grains are again visible in the diffusion layer. The nitrided layer reached a depth of $253 \pm 12$ µm. The CrN coating reached a thickness of 2–3 µm.

*3.2. Mechanical Properties*

The hardness of the hard chromium coating, the CrN coating itself and the PN + CrN duplex system is shown in Figure 5a.

The hardness values of the coatings are compared with the hardness of the base material after refining and the base material after plasma nitriding. As standard, the coatings prepared by PVD method reach depths of around 3 µm. In general, when measuring thin coatings, the indenter impression must not exceed more than 10% of the coating depth (hardness was evaluated by load HV1). This means that in the case of the coated samples, the coating + substrate system was measured. It is clear from the results that hard chromium plating achieved the same hardness as in the case of refined steel (716 HV1). The material with a deposited PVD CrN coating was found to have a decrease in surface hardness (625 HV1), which is caused by deposition at 430 °C for 6 h and thus by further thermal influence of the base material (tempering), respectively, by decreasing hardness. In this case, therefore, the hard CrN coating is "supported" by the soft base material (substrate). Plasma-nitrided steel shows an increase in surface hardness (877 HV1), however, the highest surface hardness is present in the duplex system PN + CrN (971 HV1).

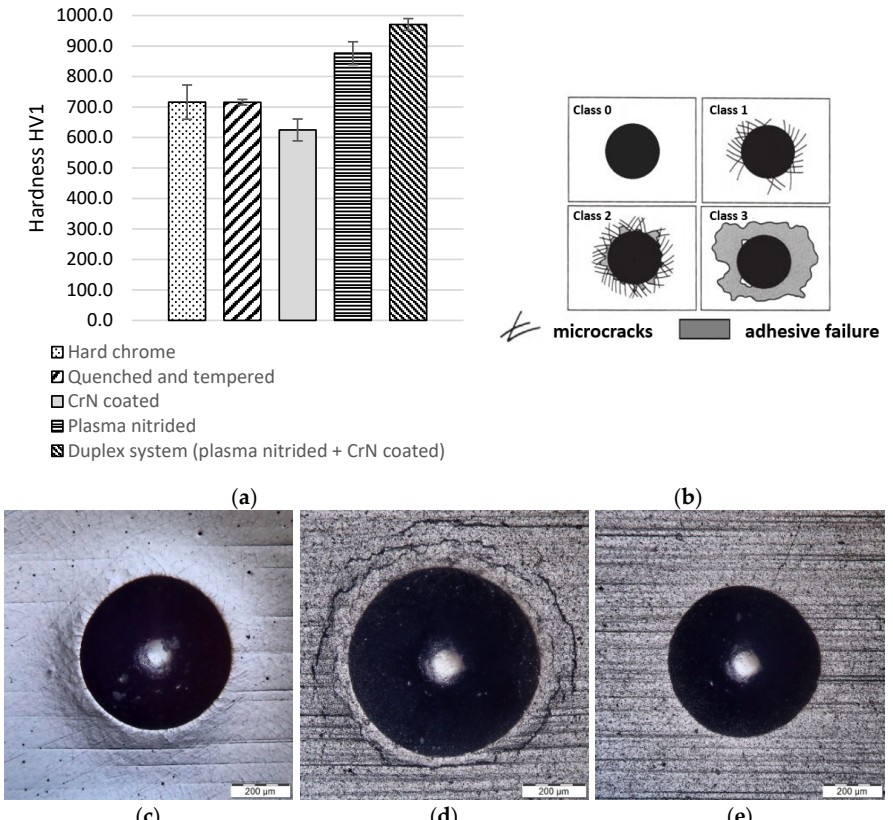

**Figure 5.** (**a**) Surface hardness; (**b**) rating scale of the indentation adhesion test (Class 0 is no cracking and no adhesive delamination, Class 1 is cracking without adhesive delamination of the coating, Class 2 is partial adhesive delamination, with or without cracking, Class 3 is complete adhesive delamination. Class observation according to ISO 26443). Impression locations: (**c**) hard chrome; (**d**) CrN coating; (**e**) PN + CrN duplex system.

A Rockwell cone, with an apex angle of 120° and a spherical rounding of the tip with a radius of 200 μm (Rockwell C), was used as an indenter in the indentation adhesion test (Mercedes test). The indenter penetrated the surface of the tested samples with a gradually increasing load up to 150 kg. This test, which is defined by ISO 26443 [48], is intended for coatings with thicknesses not exceeding 5 μm, deposited on substrates with a minimum hardness of 54 HRC [52]. The impression site was microscopically documented. The adhesive-cohesive behavior of the coating is qualitatively evaluated by comparing the appearance of the impression with the scale shown in Figure 5b. Coating damage corresponding to grades Class 0, where no damage is evident, and Class 1, where only cohesive failure in the form of cracks can be observed, are considered acceptable in terms of adhesion.

Figure 5c–e documents the impression locations of the evaluated coatings on 42CrMo4 steel, at 20× magnification.

No damage to the coating is evident on the sample with the hard chrome coating, plastic deformations of the impression edges can be noticed. The measurement is affected by the greater coating thickness (almost twice than is the recommendation in the standard). The degree of adhesion of the hard chrome coating in this case is Class 0. The CrN coating on the base material (substrate) shows several cracks around the impression site. There is no suspicion of delamination of the coating, the degree of adhesion was evaluated as Class 1. The PN + CrN duplex system shows a similar test result as the hard chrome coating. There were no cracks around the impression site, the degree of adhesion can again be assessed by the value of Class 0.

Another method used to evaluate the adhesive-cohesive behavior of coatings is the scratch adhesion test known as the "scratch test". The test is defined by ASTM C1624-05 [49] and is primarily intended for testing hard coatings with thicknesses not exceeding 30 μm. A Rockwell C type indenter loaded with a linearly increasing force of 20–80 N was used in the test. The created scratches were then evaluated microscopically. The evaluation consisted in determining the critical loads referred to as LCN, where the index N is replaced by a symbol corresponding to the areas of occurrence of specific damage to the coating [53]. The areas with coating damage and the indication of the value of the indenter loading force *Fz* are documented in Figure 6.

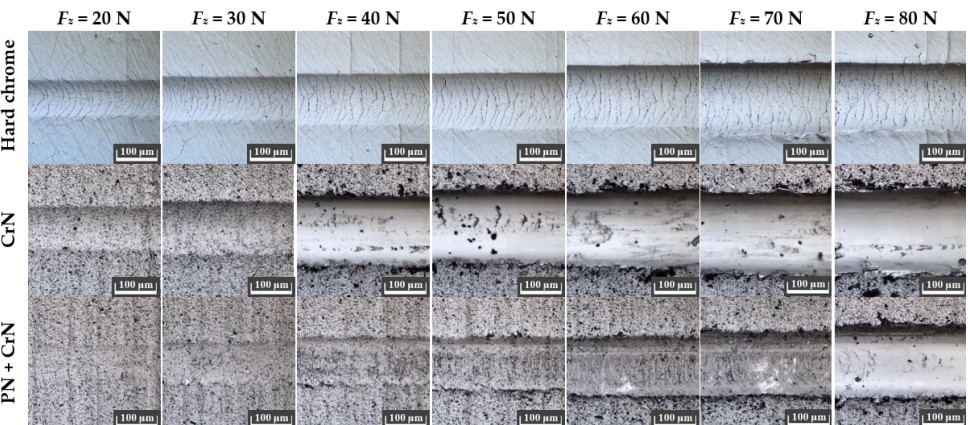

**Figure 6.** Scratches after scratch test.

From the results of the scratch adhesion test shown in Figure 6, it can be seen that cohesive defects of the hard chromium coating occurred at 20 N. Cohesive defects of the coatings were not observed for the CrN coating alone and the PN + CrN duplex system. The loads for which adhesive defects have been identified are 70 N for hard chromium coating, 40 N CrN coating and 70 N for PN + CrN duplex system. In the case of the CrN coating, the substrate was exposed at a load of 40 N (coating abrasion), this phenomenon is also visible in the case of the PN + CrN duplex system, at a load of 70 N. The substrate was not exposed in case of the hard chrome coating due to its threefold thickness compared to CrN coatings.

The evaluated samples were further subjected to an instrumented penetration test. By standard methods of measuring hardness, we are able to find out only information about the resulting plastic deformation of the surface, which can be observed only after the indenter is relieved in the form of its imprint [47]. However, the instrumented penetration test defined by ISO 14577-1:2015 [47] allows for the continuous monitoring of the vertical movement of the indenter depending on the loading force during the hardness measurement, thus creating an indentation curve. Information on plastic and elastic deformation can be obtained from this curve. This method was first introduced in this form by Oliver and Pharr in 1992 [54]. The plastic deformation work of the penetration process ($W_{plast}$), the elastic return (elastic) work of the penetration process ($W_{elast}$) and the indentation hardness ($H_{IT}$) were evaluated for the samples.

The indentation hardness can be determined from the relationship [47]:

$$H_{IT} = \frac{F_{max}}{A_p(h_c)}, \tag{4}$$

$$h_c = h_{max} - \varepsilon_c \times (h_{max} - h_r), \tag{5}$$

where $H_{IT}$ is indentation hardness, $F_{max}$ is maximum test load, $A_p(h_c)$ is projection of the contact surface of the penetrating body at a distance $h_c$, $h_r$ is the intersection of the tangent of the relief curve and the axis of the indentation depth, $h_c$ is depth of contact with the test

specimen at $F_{max}$ and $\varepsilon_c$ is correction factor (for Vickers pyramid $\varepsilon_c = 3/4$). However, it should be emphasized that the stated relations for the calculation of indenter areas only apply to indentation depths > 6 μm [47]. Due to the achieved CrN coating thicknesses, the base material (substrate) and coating system was measured for samples with CrN coating and PN + CrN duplex system. The indenter load was chosen to be 50 N, 100 N, 200 N and 400 N.

The plastic deformation work of the penetration process $W_{plast}$ of the evaluated surface treatments is graphically shown in Figure 7a. Significant changes in $W_{plast}$ are visible at 200 N and 400 N loads. Higher values were recorded for the CrN coating, while the PN + CrN duplex system showed the lowest $W_{plast}$ values. The $W_{plast}$ value of hard chrome, which is close to the $W_{plast}$ value of the ground surface, is approximately between the values obtained for the CrN coating and the PN + CrN duplex system. It is clear from the results that the hard chrome layer approaches the ground material in terms of plasticity. The CrN coating has the highest $W_{plast}$ value. The reason is the higher toughness of the base material, compared to the ground surface, caused by secondary tempering during the deposition of the PVD coating. It is necessary to realize that at higher loads, the coating + base material (substrate) system is evaluated. The lowest value of $W_{plast}$ in the duplex system PN + CrN is given by the higher hardness of the nitrided layer formed on the surface of the base material (substrate) by the nitriding process. The hard nitrided layer here acts as a "support" for the CrN coating and reduces the plasticity of the PN + CrN duplex system.

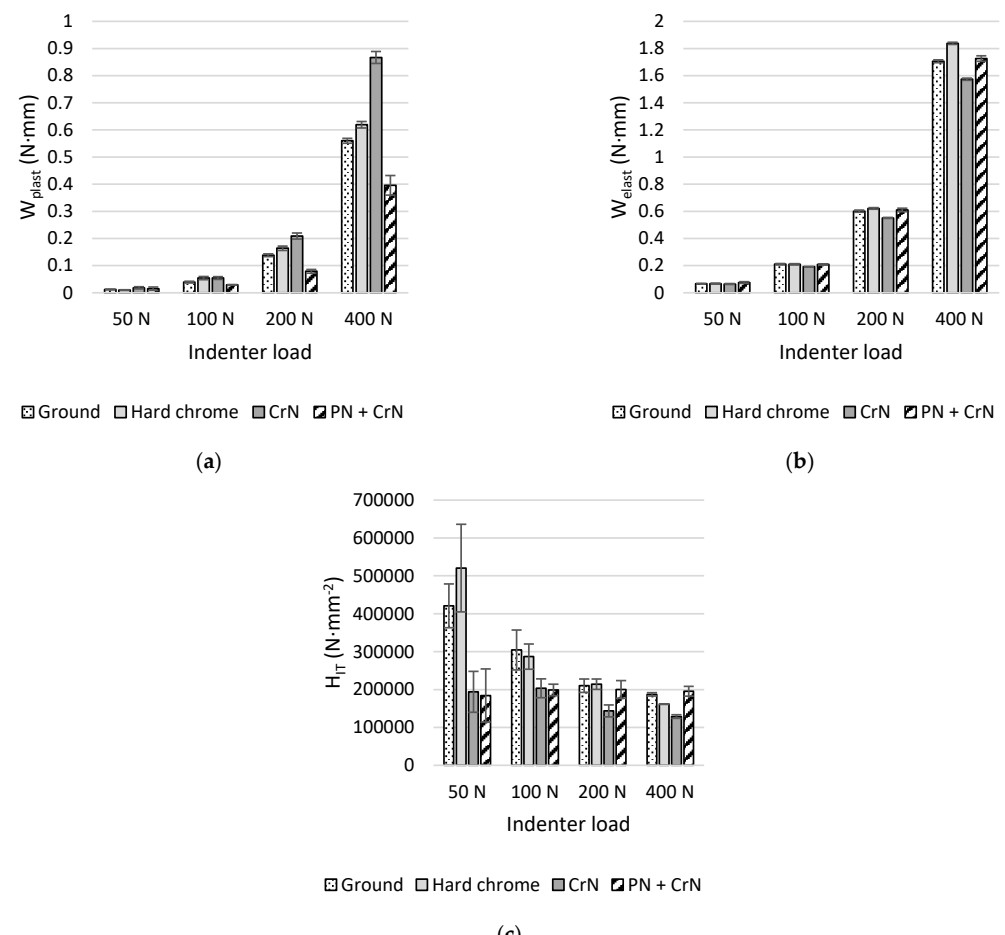

**Figure 7.** (**a**) Plastic deformation work of the penetration process; (**b**) elastic deformation work of the penetration process; (**c**) indentation hardness.

The elastic deformation work $W_{elast}$ of the penetration process is documented in Figure 7b. At lower loads of 50 N and 100 N, similar $W_{elast}$ values can be observed for all evaluated surfaces. At a load of 200 N, with the exception of the CrN coating, the

$W_{elast}$ values of the evaluated surfaces are almost identical. The loading force of 400 N led to the highest $W_{elast}$ value for hard chrome, the $W_{elast}$ value of the PN + CrN duplex system is comparable to the ground surface, the lowest $W_{elast}$ value was measured for the CrN coating.

The indentation hardness $H_{IT}$ (Figure 7c) corresponds to the degree of resistance to permanent deformation. At an indenter load of 50 N, the hard chrome coating shows the highest $H_{IT}$ value, a higher value was also measured for the ground surface. The CrN coated surface and the PN + CrN duplex system experienced a significant decrease in values $H_{IT}$. At a load of 100 N, this difference decreases. At higher loads, the influence of the base material is manifested and the $H_{IT}$ values are equalized, with the exception of the CrN coating, where a decrease in $H_{IT}$ values can be observed at loads of 200 N and 400 N. The decrease in $H_{IT}$ is also noticeable in the case of a hard chrome coating under a load of 400 N. The results show that the hard chromium coating has the highest resistance to permanent deformation at lower loads. As the load increases, the resistance to permanent deformation of the chromium coating decreases whereas the duplex system PN + CrN shows a slight increase of this resistance at 100 N load. At higher loads, the PN + CrN duplex system maintains an almost stable resistance to permanent deformation ($H_{IT}$ value).

### 3.3. Tribological Properties

The Ball-on-Flat method was used in the analysis of tribological properties of the evaluated surfaces. The load on the ball indenter was chosen as 10 N. The diameter of the ball indenter was chosen as 6.35 mm, and the material of the ball was WC. The test frequency was 5 Hz and test time 1000 s, reciprocal movement was performed on a track of 10 mm. Figure 8 shows the results after the Ball-on-Flat test. The results show that in the case of the hard chrome coating, the base material (substrate) was exposed throughout. There are visible areas of adhesive coating failure in the trace. The base material (substrate) was also exposed in the case of the CrN coating, although the wear width is halved, compared to hard chromium. Even in this case, an adhesive failure of the coating can be observed. The PN + CrN duplex system has the lowest wear rate. The base material (substrate) has only been exposed locally in a few places and the coating is capable of even further functions in this state. In terms of tribological properties, the PN + CrN duplex system achieved the best results.

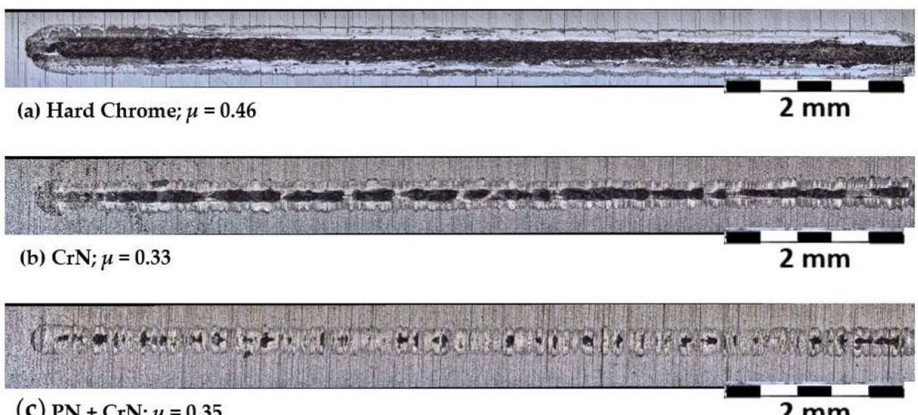

**(a) Hard Chrome; $\mu$ = 0.46**     2 mm

**(b) CrN; $\mu$ = 0.33**     2 mm

**(c) PN + CrN; $\mu$ = 0.35**     2 mm

**Figure 8.** Ball-on-Flat. (**a**) is Hard chrome, (**b**) is CrN coating and (**c**) is duplex system PN + CrN.

When applying the Ball-on-Flat method, the coefficient of friction $\mu$ was also measured. The values of the coefficient of friction $\mu$ for the evaluated surfaces are given in Figure 8. The values of the coefficient of friction $\mu$ for the CrN coating and the PN + CrN duplex system reached almost the same values. The hard chrome coating showed a higher coefficient of friction $\mu$. Due to the expected occurrence of simultaneous wear of the sample and the indenter, the transverse track profiles obtained by the inductive (touch) sensor on the

Talysurf CLI 1000 profilometer (Tylor Hobson, Leicester, UK) were used to evaluate the wear. Ten profiles were measured on Ball-on-Flat wear traces. The distance between the individual profiles was 1 mm. From these ten profiles, the middle profile of the wear trace was selected (Figure 9), which was analyzed.

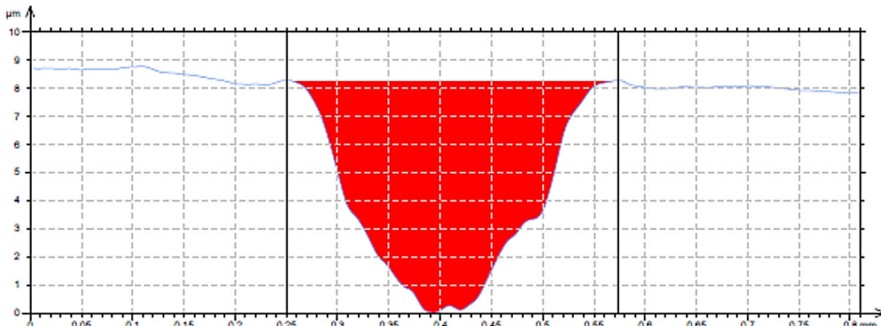

**Figure 9.** Middle profile of trace of wear—hard chrome. Maximum depth is 8.29 μm, area of the hole (red color) is 1462 μm$^2$.

The results of the measurement of wear traces of the evaluated surface treatments are documented in Table 3. The wear trace parameters, in particular the maximum trace depth and the trace area, correlate with the visual results of the wear traces in Figure 8. The widths of the wear traces were similar for the CrN coating and the PN + CrN duplex system, the hard chrome recorded a larger track width. The PN + CrN duplex system recorded a significantly smaller depth of the wear trace, compared with CrN coating and especially with hard chrome, which was reflected in the small value of area of the hole.

**Table 3.** Parameters of the wear traces.

| Coating | Maximum Depth (μm) | Width (μm) | Area of the Hole (μm$^2$) |
|---|---|---|---|
| Hard chrome | 8.29 ± 0.11 | 396 ± 22 | 1462 ± 96 |
| CrN | 2.22 ± 0.14 | 323 ± 16 | 259 ± 36 |
| PN + CrN | 0.57 ± 0.02 | 313 ± 18 | 71 ± 11 |

*3.4. Morphology and Surface Texture*

Because the functional properties of surfaces, which also include tribological properties, are based on the condition of the surface, respectively, its morphology and texture, an evaluation of these parameters was made. Tribological properties, together with the evaluation of the surface morphology and its texture, are of the greatest importance on the outer cylindrical surfaces of the gas piston, which are stressed by wear, when moving in the gas cylinder. Faster wear of the contact surfaces can, in extreme cases, lead to a reduction in gas pressure (blowing around the piston) and a reduction in the operation (functionality) of the weapon.

The formed coatings were visually evaluated using SEM (Figure 10). Surface cracks in the coating are visible on the hard chrome coated surface. Due to capillary condensation, the network of these cracks can cause the condensate to spread through imperfections and microcracks across the coating and ultimately allow the aggressive corrosive environment to penetrate to the surface of the base material (substrate). With a gas piston, the formation of condensate can be caused by the temperature difference between the gas cylinder and the surrounding environment, moisture penetration when using the weapon, poor storage, etc. After PVD deposition of the CrN coating, a morphologically fragmented surface with a structure resembling an orange peel was formed. The white spherical particles on the surface of the coating are defects such as pinholes, nodular defects and clusters of Cr ions [55]. The same morphology of the CrN coating is present in the case of the PN + CrN duplex system. Cr ion clusters are also present here.

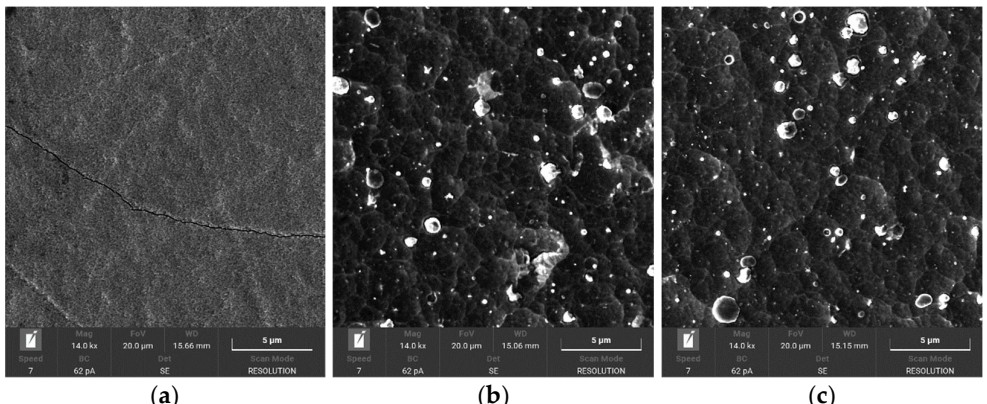

(a)            (b)            (c)

**Figure 10.** Surface morphology: (**a**) hard chrome; (**b**) CrN; (**c**) PN + CrN.

The evaluation of the surface texture consisted of the evaluation of selected 3D surface parameters and 2D roughness profile parameters. The height parameters Sa and Sz and their 2D equivalents Ra and Rz were selected from the 3D surface parameters. Two-dimensional parameters were further supplemented by Rk parameters (also known as material ratio parameters Rk), Rk, Rpk and Rvk. Selected 3D and 2D parameters, together with their name and unit, are listed in Table 4.

**Table 4.** Surface texture parameters [56].

| Parameter | Name | Unit |
|---|---|---|
| 3D amplitude parameters | | |
| Sa | Arithmetic mean deviation | μm |
| Sz | Ten-point height | μm |
| 2D amplitude parameters | | |
| Ra | Arithmetic mean deviation of the roughness profile | μm |
| Rz | Maximum height of roughness profile | μm |
| Rk parameters (parameters of material ratio) | | |
| Rk | Kernel roughness depth | μm |
| Rpk | Reduced peak height | μm |
| Rvk | Reduced valley depth | μm |

The texture of the surface, together with the evaluated 3D parameters of the surface is shown in Figure 11.

The ground surface has a directed texture with traces of the tool in the grinding direction. The reduction in the 3D surface parameters can be observed in the coating of hard chrome, which also shows a different character of the texture with three dominant elements in the shape of waves. The same values of 3D surface parameters were achieved for the CrN coating as for the ground surface. The texture also shows traces of grinding, but is more rugged. The same surface texture can be observed in the case of the duplex system PN + CrN, but there was an increase in 3D surface parameters. The increase is caused by the plasma nitriding process, which in the case of 42CrMo4 steel leads to a deterioration of the surface texture (roughness) [57]. Thus, it is obvious that in terms of the achieved height 3D parameters of the surface, the coating of hard chrome is the most suitable; the CrN coating did not show changes and the largest change was observed in the duplex system PN + CrN.

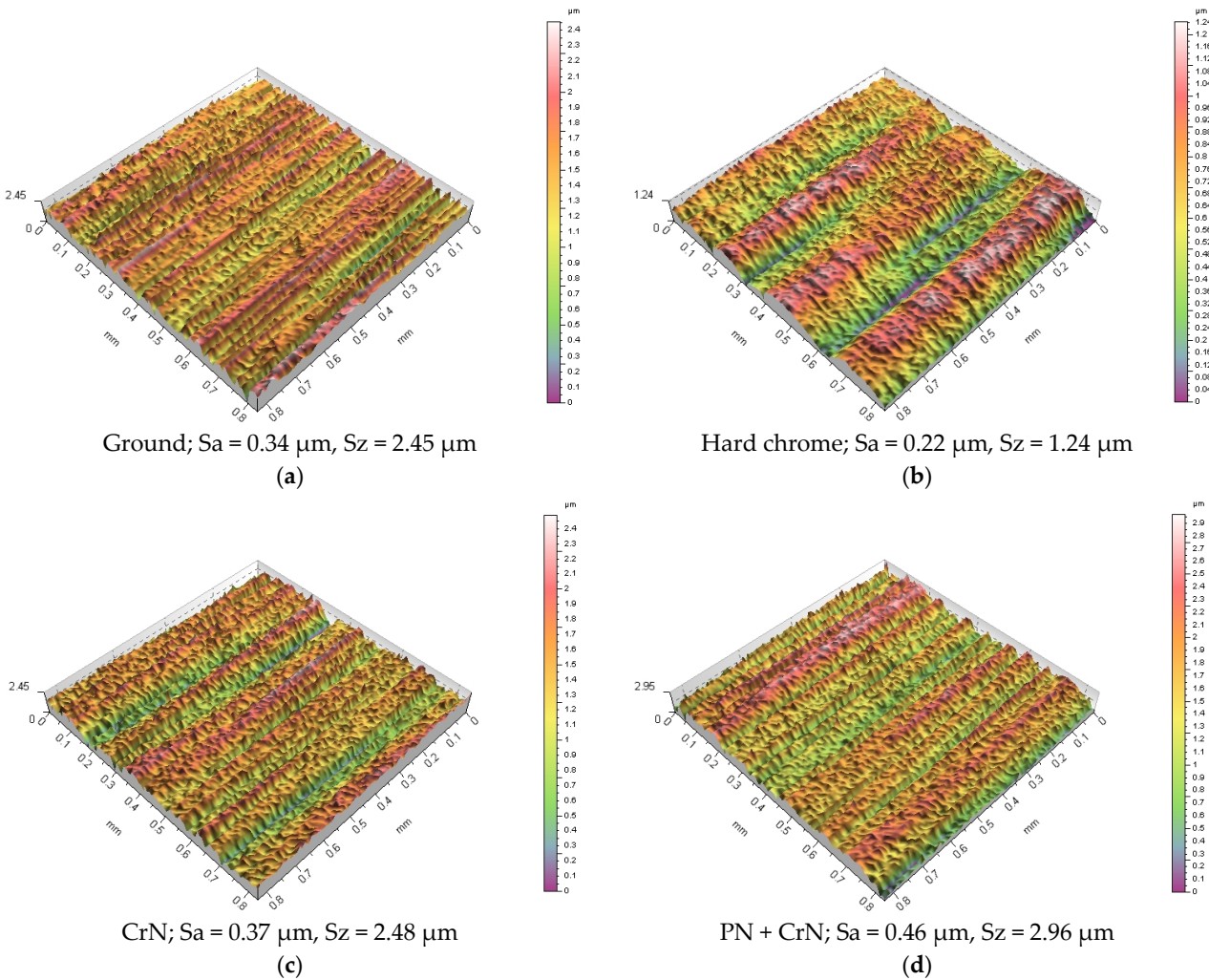

Ground; Sa = 0.34 μm, Sz = 2.45 μm

(**a**)

Hard chrome; Sa = 0.22 μm, Sz = 1.24 μm

(**b**)

CrN; Sa = 0.37 μm, Sz = 2.48 μm

(**c**)

PN + CrN; Sa = 0.46 μm, Sz = 2.96 μm

(**d**)

**Figure 11.** Surface texture and 3D parameters. (**a**) is ground surface, (**b**) is hard chrome coating, (**c**) is CrN coating and (**d**) is duplex system PN + CrN.

The evaluation of 2D surface roughness profiles is documented in Figure 12. The surface roughness profiles correlate with the textures shown in Figure 11. The ground surface has a non-periodic profile with a large number of sharp peaks and valleys. A completely different microgeometry of the roughness profile can be seen in hard chrome, there are visible dominant protrusions with a larger radius of rounding of the peaks and wider valleys. The roughness profile of the CrN coated surface can be characterized as random, however, in comparison with the ground surface, dominant peaks and wider valleys are visible. In the PN + CrN duplex system, the similarity of the surface microgeometry with the microgeometry achieved with the CrN coating is obvious, but the profile elements (peaks and valleys) are larger.

The values of the height parameters of the roughness Ra and Rz correlate with the 3D height parameters of the area Sa and Sz. This means that the smallest values, compared to the ground surface, showed a coating of hard chrome and the largest change in the form of increased parameters was recorded by the duplex system PN + CrN. When evaluating the Rk parameters (material ratio), it is clear that the hard chrome coating exhibits the most suitable properties. The Rk parameter has the smallest value, the surface is flatter and more load-bearing and will wear more slowly. The lowest Rpk value also corresponds to slow wear. Hard chromium also reached the lowest value of Rvk, which means a small depth of the valleys. The increase in the parameters Rk and Rpk, in comparison with the ground surface, can be observed for the CrN coating, while the parameter Rvk decreased. The

surface is therefore more rugged, with greater depth of the roughness kernel and higher peaks, which will wear faster. The largest changes in the parameters Rk and Rpk were achieved in the duplex system PN + CrN, in the form of their increase. The Rvk parameter is the same as in case of the ground surface. The surface with the PN + CrN duplex system therefore has the greatest depth of the roughness kernel and the height of the peaks, so it should be the least load-bearing, rugged and wear the fastest.

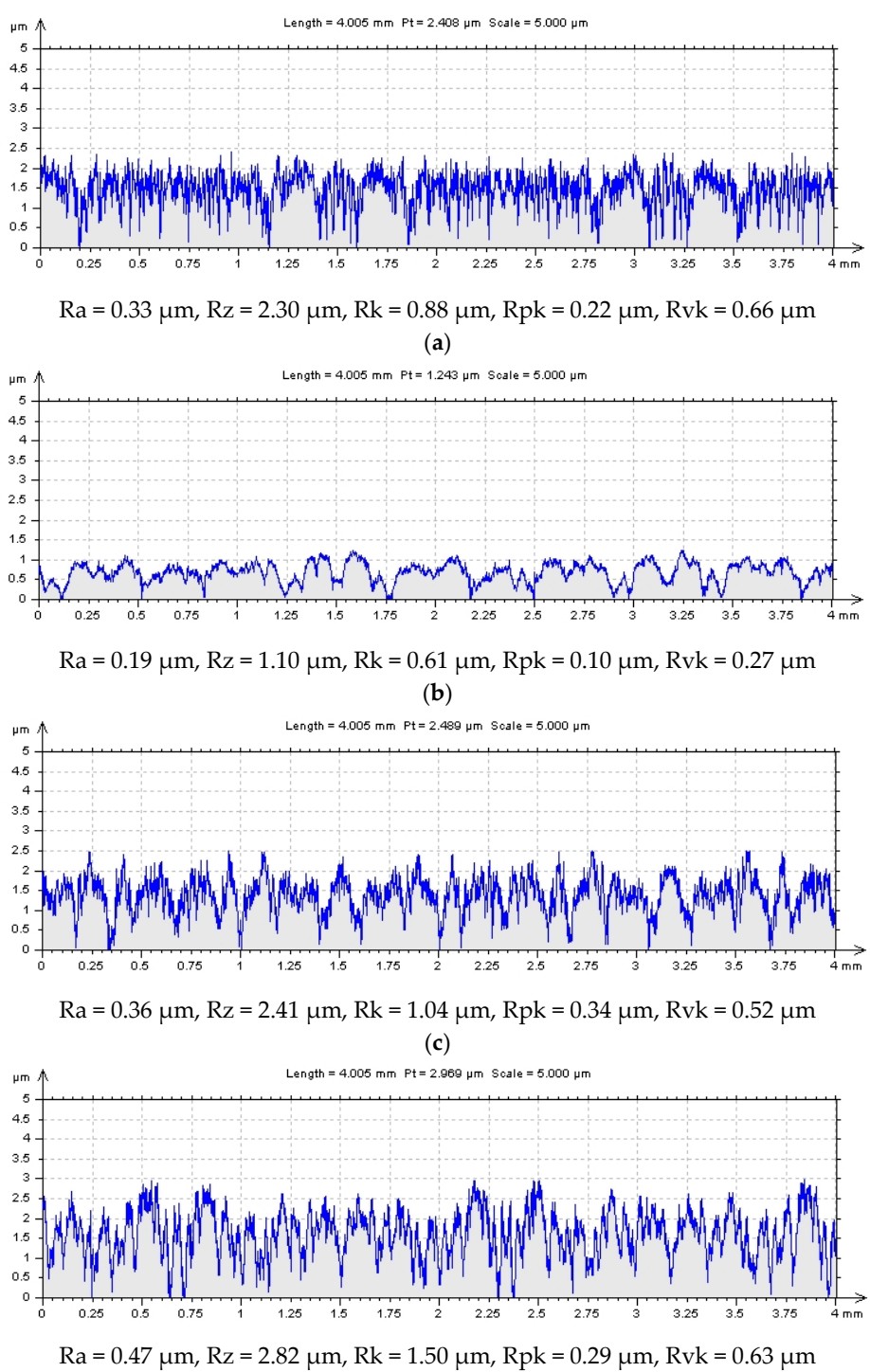

Ra = 0.33 μm, Rz = 2.30 μm, Rk = 0.88 μm, Rpk = 0.22 μm, Rvk = 0.66 μm

(**a**)

Ra = 0.19 μm, Rz = 1.10 μm, Rk = 0.61 μm, Rpk = 0.10 μm, Rvk = 0.27 μm

(**b**)

Ra = 0.36 μm, Rz = 2.41 μm, Rk = 1.04 μm, Rpk = 0.34 μm, Rvk = 0.52 μm

(**c**)

Ra = 0.47 μm, Rz = 2.82 μm, Rk = 1.50 μm, Rpk = 0.29 μm, Rvk = 0.63 μm

(**d**)

**Figure 12.** Surface roughness profiles with 2D parameters: (**a**) ground; (**b**) hard chrome; (**c**) CrN; (**d**) PN + CrN.

However, it is necessary to state that the characteristics of individual 3D surface parameters and 2D roughness profile parameters are mainly related to machined surfaces. For surfaces with coatings or diffusion layers, namely duplex systems layer + coating, the behavior of the functional surface under load can be completely different, which is evident from the evaluation of tribological properties. The same conclusions were reached in [58,59], where the nitrided surfaces of 42CrMo4 steel were evaluated. Despite the deteriorated surface roughness after the plasma nitriding process, better tribological surface properties were achieved (lower coefficient of friction $\mu$ and wear coefficient $K$). Similar results were achieved at work [60,61].

## 4. Conclusions

In summary, the application of the PN + CrN duplex system had a higher hardness than hard chrome. The hardness of the CrN coating (measured as the hardness of the base material + coating system) was lower than that of hard chrome. The PN + CrN duplex system showed the same adhesive properties as hard chrome and better than the CrN coating itself. The PN + CrN duplex system and the CrN coating did not show cohesive defects. Hard chrome showed cohesive defects already at a load of 20 N. The CrN coating reached the highest value of plastic deformation work $W_{plast}$ (it should be noted that the CrN + base material system was measured). The PN + CrN duplex system reached the lowest $W_{plast}$ value. Hard chrome reached the highest value of elastic deformation work $W_{elast}$. The lowest $W_{elast}$ value was observed for the CrN coating. The PN + CrN duplex system achieved the same $H_{IT}$ indentation hardness as hard chrome. The $H_{IT}$ of the CrN coating was lower. The PN + CrN duplex system achieved the best tribological properties (Ball-on-Flat method) and the PN + CrN duplex system and the CrN coating achieved a lower coefficient of friction $\mu$ than hard chrome. The PN + CrN duplex system and the CrN coating have a more rugged surface morphology than hard chrome. The surface texture parameters (3D surface parameters Sa and Sz) of hard chrome have the lowest values. The highest values were recorded by the PN + CrN duplex system. Hard chrome has the lowest surface roughness while the PN + CrN duplex system has the highest roughness.

The results show that the PN + CrN duplex system can be considered an acceptable replacement for the hard chrome coating in terms of mechanical and tribological qualities. The CrN coating on the base material (substrate) has lower mechanical and adhesive capabilities, making it less appropriate as a hard chrome substitute. The impact of lower roughness parameters on the functional behavior of the gas piston during weapon operation can only be determined by conducting functional testing with real fire, although the PN + CrN duplex system is projected to achieve similar parameters as the hard chrome coating.

**Author Contributions:** Conceptualization, D.D. and Z.P.; Methodology, R.V.; Software, J.P.; Validation, Z.S., J.S. and M.S.; Formal analysis, D.D.; Investigation, R.V.; Resources, Z.J. and S.K.; Data curation, Z.P.; Writing—original draft, D.D. and R.V.; Writing—review & editing, Z.P.; Visualization, J.S.; Supervision, Z.S.; Project administration, M.S.; Funding acquisition, J.P. All authors have read and agreed to the published version of the manuscript.

**Funding:** This research was supported by the specific research project 2020 SV20-216 at the Department of Mechanical Engineering, University of Defence in Brno, by the "Project for the Development of the Organization DZRO VAROPS" and by the project with the grant Modern technologies for processing advanced materials used for interdisciplinary applications FSI-S-22-7957.

**Institutional Review Board Statement:** Not applicable.

**Informed Consent Statement:** Not applicable.

**Data Availability Statement:** Not applicable.

**Acknowledgments:** We thank the Faculty of Military Technology, University of Defense in Brno and the Faculty of Mechanical Engineering, Brno University of Technology, for the support provided during the research.

**Conflicts of Interest:** The authors declare no conflict of interest.

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
