# Peer review of "Possibilities of Using the Duplex System Plasma Nitriding + CrN Coating for Special Components"

_coatings, doi:10.3390/coatings12121953_

Round 1
Reviewer 1 Report
The authors investigated the mechanical and tribological properties of CrN coating deposited by PVD technology on a substrate in the form of 42CrMo4 steel, which is widely used for the production of exposed weapon components, and also the properties of both the CrN coating itself and the duplex system plasma nitriding + deposition of the CrN coating (PN + CrN). The PN + CrN duplex system was harder than hard chrome, and had better tribological qualities (lower friction coefficient), but had a rougher surface texture. The PN + CrN duplex system has proven to be an acceptable alternative to hard chrome coating for exposed components of small-caliber weapons, which can be used in the manufacturing of arms.
I consider that the results of the presented study may be of interest to readers of your journal. Additionally, the abstract of the manuscript, introduction, and conclusions was good, and the results were adequately discussed. But I have some criticisms about the article below. After these corrections are made by the Authors, the manuscript should be accepted for publishing in the Coatings.
My corrections:
1. Some recent papers about the tribological properties of some steels should be added to the introduction part or other parts of the manuscript. For examples :
i) https://doi.org/10.1515/rams-2020-0030
ii) DOI: 10.12693/APhysPolA.132.455
iii) https://doi.org/10.1007/s11665-020-04796-9
Author Response
Dear Reviewer,
thank you for your comments on the article. Based on your comments, I added 2 resources recommended by you to the article.
Reviewer 2 Report
In this manuscript, hard chrome coating, self-deposited CrN coating and PN + CrN duplex system were prepared and their metallography, mechanical and tribological properties and surface textures were analyzed and compared. The PN + CrN duplex system showed higher hardness than hard chrome, better tribological properties, but worse surface texture, indicating that it is a suitable alternative to the hard chrome coating. However, some issues should be addressed before publication.
1. The manuscript is more likely a report, not a paper. Some places need to be streamlined, especially line 446-485.
2. The white spherical particles on the surface of the coating are clusters of Cr ions. Can the authors provide sufficient evidence? Is EDS OK?
3. “The values of the coefficient of friction μ for the evaluated surfaces are given in Fig. 8.” “The texture of the surface, together with the evaluated 3D parameters of the surface is shown in Fig. 12.” Practically, friction μ and 3D parameters are not shown in the Figures, but in Captions. It is unsuitable. The authors can refer their Fig. 13 which is mistakenly typed as 12.
4. The authors attributed the measured hardness of the CrN coating to the hardness of the base material + coating system. Why did not the authors increase the thickness of CrN coating so that the measured hardness is the hardness of the CrN coating?
5. Fig. 10 should be deleted.
6. The terminology should preferably be consistent, such as “indentation hardness” and “impression hardness”.
7. The magnification is inappropriate to appear in the Captions, because images are sometimes scaled. The scale bar is preferred.
8. They are some typos, such as “42CrMo2”, “CrO3”, “μlap” line 704. etc.
Author Response
Dear Reviewer,
thank you for your valuable comments on the article. Based on your comments, I made the following corrections:
ad 1) I significantly shortened lines 446-485 and deleted the description of individual surface roughness parameters;
ad 2) The description of spherical particles on the surface has been corrected. We have the results from the EDS and the chemical composition of the spherical particles according to the EDS analysis is approx. 80 wt. % Cr and approx. 20 wt. % Fe (see Annex);
ad 3) Your comments have been incorporated;
ad 4) the thicknesses of the coatings deposited by the PVD method range up to 5 - 6 µm. Greater thickness of the coating is uneconomical and, especially from the point of view of mechanical properties, such coatings would be unusable. For this reason, when measuring microhardness, the coating+substrate system is always measured. If we would like to measure only the hardness of the coating, we have to use the nanohardness measurement method. However, by measuring the nano-hardness, we are not able to affect the behavior of the coating+substrate system under real loads in operation;
ad 5) Figure 10 was deleted;
ad 6) Your comments have been incorporated;
ad 7) Your comments have been incorporated;
ad 8) Your comments have been incorporated. The designation of chromium oxide CrO3 is not a typo, this oxide is used for hard chrome plating. I did not find an error on line 704, which referred to a literary source.

Reviewer 3 Report
The author could discuss how the texture (grain orientations) affect the coating hardness.
Author Response
Dear Reviewer,
thank you for your comment. Your suggestion is very interesting and expands the field of this issue by solving an interesting problem. However, this issue was not addressed in the presented article. The article has the character of a practical application of the proposed solution into industrial practice, and the significance of the microstructure of the CrN coating was not addressed. For the deposition of the PVD coating, standard coating parameters used in industry were used. In particular, adhesion parameters and hardness were evaluated.
Reviewer 4 Report
The authors have discussed the application of PN+CrN duplex coating as a replacement for hard chrome coating in a gas piston of a small bore weapon. The flow is nice, the problem is properly addressed, experimentation is adequate and results are in support. The topic of interest and recommended for publication with the following minor changes:
1. line 207: what is the 'hardness of K'? what is the value?
2. line 317: the adhesion of CrN seems to be HF4 rather than HF2.
3. Figure 7c: The resistance to permanent deformation remains almost constant for PN+CrN coating with increasing load (50 -400 N), but why does the same decrease for CrN coating? In fact, the CrN coating shows the lowest HIT value at 400N. Does it indicate that nitriding the surface decreases the coating+substrate system's indentation hardness? Why?
4. Figure 8 and Table 2: What is the reason for PN+CrN coating showing significantly low wear but slightly higher friction than CrN coating? Is this number indicating average friction?
5. Scales are not visible in Figure 8, can you make it clear?
6. Figure 12: It would be more meaningful to readers if the scales are kept the same for all four figures rather than using different ones.
7. Images of 2D surface roughness profiles are actually Figure 13, correct the figure number in the image.
8. General comments: grammatical mistakes are there and some sentences need reconstruction, lines 395-397 for example.
Author Response
Dear reviewer,
thank you for a number of valuable comments. I have the following comments regarding your comments:
ad 1) The "K" hardness shown on line 207 is used to mark grinding wheels from various manufacturers. This number has no unit and does not define the hardness of the wheel, but its ability to hold the grains in bond is not related to the hardness of the grains. Based on this, grinding wheels are divided into soft, medium and hard;
ad 2) line 317, the adhesion of the CrN coating is really HF2, because with HF4 there would have to be delamination of the coating from the substrate in the cracked areas, i.e. the substrate would be visible around the impression;
ad 3) Fig. 7c - yes, the PN+CrN system maintains a constant almost constant resistance to permanent deformation under the selected load, which is required. The decrease when only the CrN coating is used is due to the large hardness gradient between the coating and the substrate. For the PN+CrN system, the hardness drop gradient is significantly smaller because the CrN coating is "supported" by a hard nitrided layer. The PN+CrN system does not reduce the indentation hardness, which is evident from the graph. The values are similar to those of the CrN coating at lower loads. At lower loads, it can be said that only the hardness of the coating is measured. At higher loads, the substrate also plays a role (we measure both the coating and the substrate), which was manifested at a load of 400 N and a decrease in the hardness of the CrN coating on the ground surface;
ad 5) Corrected;
ad 6) Images have been enlarged and scales have been removed;
ad 7) Corrected;
ad 8) The sentences on lines 395-397 have been reworded.
Reviewer 5 Report
The paper presents an interesting approach based on the Possibilities of Using the Duplex System Plasma Nitriding + CrN Coating for Special Components. However, the innovation of the current research work should be further highlighted and emphasized. At the same time, the authors should consider the following comments to greatly improve the quality of the paper.
1. In the abstract, kindly introduce the research problem in the first few lines.
2. The introduction needs to be improved by relating to the mechanics of the studied materials and their mechanical characteristics. The references to be included are: 10.1016/j.jiec.2022.06.023, 10.1016/j.polymertesting.2017.09.009, 10.1016/j.compstruct.2021.114698, 10.1177/0731684417727143, 10.1002/app.46770, 10.1016/j.porgcoat.2022.107015.
3. Kindly add a table that describes the main physical and chemical properties of the raw materials used in this study.
4. Were the preparation methods for the chemical heat treatment and the deposition described by the authors come in accordance with a certain standard or do they follow previous procedures?
5. What was the reason for selecting the scratch adhesion test in this work package? Also, does the specified test conditions (including the load range of 20 N – 80 N) in this research replicate any given application?
6. What is the reference used for calculating the indentation (equation 4 and 5)?
7. Kindly mention the operating parameters for the SEM scans, including the accelerating voltage and working depth.
8. The surface texture and height bar in Figure 12 aren't clear at all. Would you kindly obtain higher quality images that show the height variations and surface roughness?
9. The conclusion needs to be modified to summarize the research outcomes in short statements with clear observations.
Author Response
Dear reviewer,
thank you for your comments on the article. Regarding your comments, I am writing the following comments:
ad 1) Your comment has been processed;
ad 2) With all due respect to you, the article references you suggested do not correspond at all to the topic and focus of the article. For this reason, the articles recommended by you have not been added. If you have a problem with this point, I will ask the Editor;
ad 3) The table is not needed, all necessary information about the materials used is given in the "Results" chapter.
ad 4) The parameters of chemical-heat treatment and deposition of coatings are based on experience with these processes. Parameters that are used for highly stressed components were selected;
ad 5) The scratch test was chosen for the reason that it determines the adhesive and cohesive properties of coatings, even directly for manufacturers of coating equipment and manufacturers of tool coatings. Another reason was that our workplace is equipped for this type of tests. The load range was determined experimentally based on the results of long-term research at our workplace and based on the recommendations of companies that deposit hard coatings;
ad 6) The reference to the literature (standard) has been added;
ad 7) SEM parameters were added;
ad 8) Fig. 12 was corrected;
ad 9) In my opinion, the conclusion is clearly written and summarizes the results found.
Reviewer 6 Report
Please amend the following:
i) I would suggest improving the readability of figures 9, 12, and 13 (the numeration of the last one is wrong) via increased scale bars.
ii) How is the area of holes connected with their linear sizes? The apparent relation does not match the data presented.
iii) What is the arithmetic mean deviation? Please introduce if it is not the same as the standard mean deviation.
iv) To use (at least roughly) the equation in section 2.1, one must present the combustion reaction and the calculated amount of combustion gases. Use ms in fig. 2. How was the change of enthalpy calculated for such a dynamic combustion?
Author Response
Dear reviewer,
thank you for your comments on the article. I have the following comments on your comments:
ad 1) the readability of Figures 9, 12 and 13 has been increased, the numbering corrected and the scales enlarged;
ad 2) The figure showing the area of the wear track must be taken with a grain of salt, because the scale in the Z axis is different from the scale in the X axis. The size of the area is given by calculation in the software of the evaluation program. The wear track profile is smoothed and filtered (roughness removed). The indicated area value then corresponds to the actual area of the wear track.
ad 3) Only the standard deviation was determined as part of the evaluation of the results. Root mean square error was not evaluated because only 5 measurements were taken;
ad 4) The units in Fig. 2 have been adjusted to ms. The enthalpy calculation is given in the appendix.

Round 2
Reviewer 3 Report
Good revisions
Author Response
Dear Reviewer,
thank you for your comment on the revised article.
Reviewer 5 Report
The authors still could fulfill these comments:
2. The introduction needs to be improved by relating to the mechanics of the studied materials and their mechanical characteristics. The references to be included are: 10.1016/j.jiec.2022.06.023, 10.1016/j.polymertesting.2017.09.009, 10.1016/j.compstruct.2021.114698, 10.1177/0731684417727143, 10.1002/app.46770, 10.1016/j.porgcoat.2022.107015.
3. Kindly add a table that describes the main physical and chemical properties of the raw materials used in this study.
4. Were the preparation methods for the chemical heat treatment and the deposition described by the authors come in accordance with a certain standard or do they follow previous procedures?
9. The conclusion needs to be modified to summarize the research outcomes in short statements with clear observations.
Author Response
Dear Reviewer,
thank you for your comments on the revised article. I have processed your comments in the following way:
ad 2) I improved the introduction with your publication suggestions, but only used the 3 recommended articles;
ad 3) I have added a table with selected properties of the materials used;
ad 4) I completed the genesis of the selected parameters of the chemical-heat treatment and deposition of the coating;
ad 9) I edited the conclusion.
I ask you to revise the article. Thank you.
Round 3
Reviewer 5 Report
The article can be accepted.
Author Response
Dear Reviewer,
thank you for revising our article and accepting the changes we made in the article.